# RNA editing generates cellular subsets with diverse sequence within populations

Dewi Harjanto[1,*], Theodore Papamarkou[2,*], Chris J. Oates[3], Violeta Rayon-Estrada[1], F. Nina Papavasiliou[1] & Anastasia Papavasiliou[4]

RNA editing is a mutational mechanism that specifically alters the nucleotide content in transcribed RNA. However, editing rates vary widely, and could result from equivalent editing amongst individual cells, or represent an average of variable editing within a population. Here we present a hierarchical Bayesian model that quantifies the variance of editing rates at specific sites using RNA-seq data from both single cells, and a cognate bulk sample to distinguish between these two possibilities. The model predicts high variance for specific edited sites in murine macrophages and dendritic cells, findings that we validated experimentally by using targeted amplification of specific editable transcripts from single cells. The model also predicts changes in variance in editing rates for specific sites in dendritic cells during the course of LPS stimulation. Our data demonstrate substantial variance in editing signatures amongst single cells, supporting the notion that RNA editing generates diversity within cellular populations.

[1] Laboratory of Lymphocyte Biology, The Rockefeller University, New York, New York 10065, USA. [2] School of Mathematics and Statistics, University of Glasgow, Glasgow G12 8QW, UK. [3] School of Mathematical and Physical Sciences, University of Technology Sydney, Ultimo, New South Wales 2007, Australia. [4] Department of Statistics, University of Warwick, Coventry CV4 7AL, UK. * These authors contributed equally to this work. Correspondence and requests for materials should be addressed to F.N.P. (email: papavas@rockefeller.edu) or to A.P. (email: A.Papavasiliou@warwick.ac.uk).

The central dogma of biology assumes faithful transmission of information from DNA to RNA to protein. However, changes in DNA methylation or the chromatin state strongly affect not only the flow of information but also its heritability. In addition to these epigenetic alterations, there has been growing interest in investigating the epitranscriptome, or modifications that occur at the RNA level, which can affect both the regulation of gene expression, and what is actually being expressed by directly altering the decoding of proteins.

One type of modification of interest is RNA editing, which involves the dynamic alteration of specific nucleotides in transcribed RNA. The advent of RNA-seq technology has facilitated the identification of RNA editing events in the transcriptome, and numerous studies cataloguing such events in diverse systems have been published[1–3]. RNA editing is mediated by two types of deaminase enzymes: (1) ADARs, which convert adenosine to inosine (A to I); and (2) APOBEC1 (as well as APOBEC3A in humans, as recently described in ref. 4), which converts cytosine to uracil (C to U). RNA editing has been implicated in processes as diverse as neuronal and immune cell development and function[5,6], and oncogenesis and tumour progression[7–10]. However, the functional relevance of specific editing events, especially when taken in aggregate, is just now beginning to be explored.

Specific RNA editing events found from RNA-seq are typically presented in the literature with their detected editing rates, that is, the number of edited reads divided by the total number of reads mapped to a specific site. RNA editing rates vary widely, from $<1$ to $>90\%$ per transcript per site; in our own analyses using stringent filtering, putative C-to-U sites are edited at an average of $\sim 15-20\%$ (Supplementary Data 1). To date, most studies have focused on highly edited transcripts (for example, GLUR2 in the brain[11] and AZIN1 in cancer[12]), on the assumption that those will be most meaningful for function; however, even highly edited transcripts exist in a milieu where the vast majority of edited transcripts are altered at substantially lower levels, raising questions about the biological significance of editing in aggregate. Consequently, two hypotheses have been proposed. The first, proposed by Gommans and Maas, is that the abundance of low-frequency RNA editing events observed from bulk RNA-seq data is an accurate representation of what happens in each cell. Such low-frequency events may be 'noise', which may still fulfil a biological function as an alternative mechanism to genomic-level mutations for probing potentially advantageous adaptations[13]. The second, alternative hypothesis presented by Pullirsch and Jantsch, is that RNA editing may actually be occurring at very high rates in specific subsets of cells, serving to diversify cell populations[14].

To test these hypotheses, we sought to compare editing frequencies derived from population-based RNA-seq data with RNA-seq data from single cells. There are a number of factors that affect our ability to detect editing, including site mappability, editing frequency and coverage. RNA editing detection, especially of sites that are not highly edited, is complicated by variations in capture efficiency. This is not a concern in conventional bulk RNA-seq, which is performed using a large amount of cells or tissue, since the loss of even a large portion of the starting material may be tolerated if the remaining fraction can still provide a representative sample of the population's gene expression profile. But these sampling issues substantially impact the ability to detect editing when libraries are made from single cells. As noted by ref. 15, the cumulative losses during library preparation, primarily due to inefficiencies in the reverse transcriptase and PCR amplification steps, can severely impair detection of lowly expressed genes in single-cell RNA-seq, where one is working from extremely limited material. Transcript detection efficiency has been estimated at 20% from single cell RNA-seq[16]. Thus, if an editing event is not recovered in a single-cell data set, it can be attributed to one of two possibilities: either the site was not edited in the cell's transcriptome; or alternatively, edited transcripts mapping to that region may have been present but were not captured during the sequencing library preparation process.

Here we have compared APOBEC1-mediated C-to-U RNA-editing rates derived from single cells with 'bulk' editing rates recovered from populations of cells. To account for the stochasticity inherent in single-cell RNA-seq, we have taken a statistical approach: we have used a hierarchical Bayesian method to model the levels of variability of rates per site across single cells and in bulk samples, and to quantify the posterior variance of editing rates in single cells. Our approach reveals that while some transcripts are edited with low variance across cells (as predicted by Gommans and Maas), others exhibit high editing rate variance across cells (as hypothesized by Pullirsch and Jantsch). The existence of the latter set of transcripts supports the hypothesis that 'bulk' RNA editing represents a population average of cells that show a wide spectrum of editing rates. Our work suggests that the sequence diversity contributed by RNA editing might provide subsets of cells with distinct informational content. Further, our work implies that RNA editing might underlie or anticipate the functional heterogeneity apparent in populations of cells of monocytic lineage (for example, the macrophages or dendritic cells we have focused on here), the mechanism for which is currently not understood.

## Results

**RNA editing in populations and in single cells.** By comparing wild-type and $APOBEC1^{-/-}$ bulk RNA-seq data sets, using an in-house pipeline (Supplementary Fig. 1 and Supplementary Table 1), we detected 410 high-confidence C-to-U RNA editing events across 275 transcripts in bone marrow-derived macrophages (Supplementary Data 1). Within this terminally differentiated, homogeneous population of resting macrophages (Fig. 1a), nearly all (97%) of the C-to-U events were found to occur in 3′-untranslated regions of transcripts, similar to what was observed in small intestinal enterocytes[3]. Bioinformatically, from bulk data, we observed that C-to-U editing occurred either at a single site in a transcript (termed 'site-specific' editing) or at a set of distinct sites that were separated by a range of distances within a transcript (termed 'hyperedited') (Fig. 1b).

To explore if the bulk editing rates, representing population averages, were recapitulated on a per cell basis (Fig. 1c), we performed single-cell RNA-seq on wild-type macrophages. The cells were sequenced to an average depth of 3.8 million 100-nucleotide reads, resulting in an average of 3.5 million mapped reads (1.6 million, after deduplication) per cell. A scatterplot of the fragments per kilobase of transcript per million mapped reads values from the ensemble of reads from the 24 cells compared with the bulk experiment indicated high correlation, with a Spearman correlation coefficient of 0.884 and a Pearson correlation coefficient of 0.908 (Fig. 1d). Robustly expressed transcripts such as B2m showed little variability in expression levels in the single-cell libraries (Fig. 1e). However, that was not the case for all edited transcripts. The high confidence C-to-U sites identified from the bulk RNA-seq were covered to varying degrees across the 24 cells (Fig. 1e), with lowly and moderately expressed genes showing the most variability, consistent with previous single-cell studies[17]. However, editing variability did not necessarily correlate with APOBEC1 levels as represented by transcript per million mapped read values (Fig. 1f, bottom).

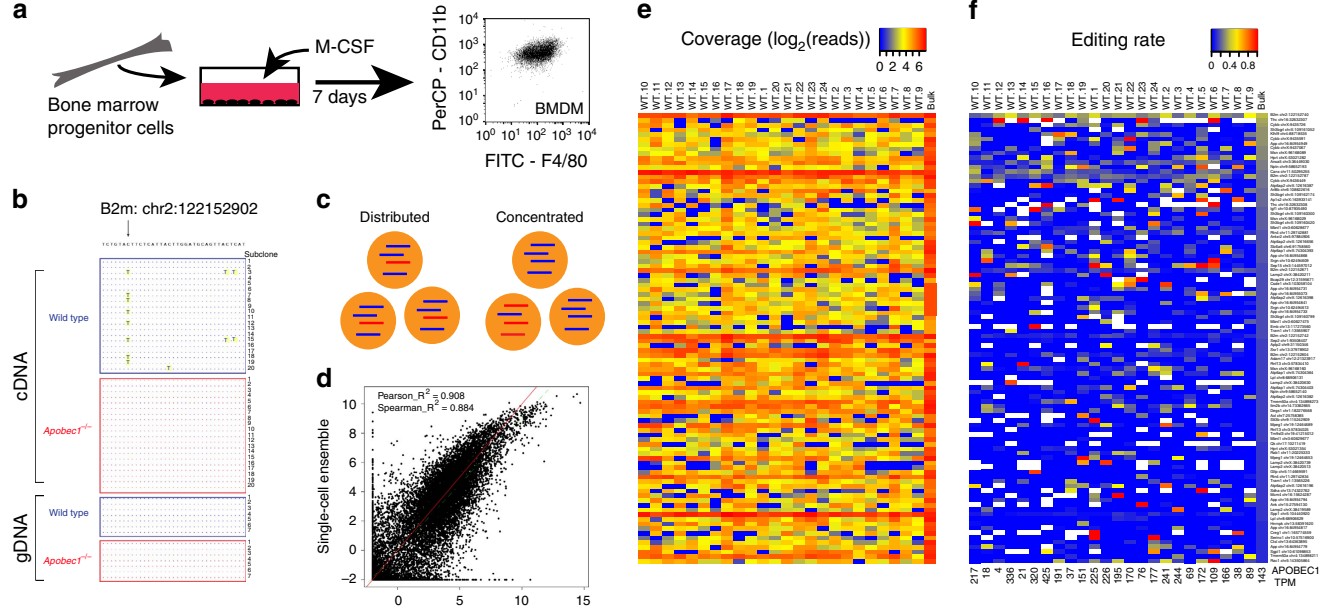

**Figure 1 | RNA editing in macrophages.** (**a**) The macrophages analysed in this study are derived from murine bone marrow, and matured *in vitro* with macrophage colony-stimulating factor (M-CSF). They are terminally differentiated (as indicated by homogeneous CD11b+ and F4/80+ staining) and have exited the cell cycle, and thus constitute a uniform population. (**b**) Example of a site-specific C-to-U editing event, on the 3′-untranslated region of B2m (chr2:122152902), with Sanger sequencing of cDNA from RNA and genomic DNA from macrophages from wild-type and *APOBEC1*$^{-/-}$ mice. (**c**) Cartoon of the two hypotheses to be tested: edited reads (red) may be either distributed proportionally across a population with respect to unedited reads (blue) corresponding to the same segment of a transcript, or concentrated in subsets of cells. In both distributed and concentrated cases, the effective bulk editing rate is 25%. (**d**) Log2(FPKM + 1) values plotted for a bulk RNA-seq experiment and for the ensemble of reads from 24 single-cell RNA-seq experiments for macrophages from wild-type mice. Red line indicates $x = y$, and dotted green line indicates the fitted regression curve. Heat maps plotting the coverage in log2(reads) (**e**) and the corresponding editing rates (**f**) for high confidence C-to-U edited sites in the 24 single-cell and bulk macrophage RNA-seq data sets. Expression levels of APOBEC1 in transcript per million mapped reads are also provided under the heat map of editing rates (and also in Supplementary Data 2). Sites are sorted in order of descending bulk editing rate. One edited site in B2m is at the top of the list; an edited site within Cybb is third from the top of the list. FPKM, fragments per kilobase of transcript per million mapped reads.

The sites that are covered generally exhibit C-to-U editing in at least one of the profiled cells (402 out of 410 sites were covered in at least one of the cells; 222 of those sites were also edited in at least one of the 24 cells), and there appears to be a wide range of editing rates for some specific sites (Fig. 1f). For instance, chrX:9436449 on the uniformly well-covered transcript Cybb is edited in 61.5% of the reads mapping to that site in cell 6 (compared to just 23% in the bulk experiment), while the same site isn't edited at all in 11 other cells where the site is covered. Similarly, chr3:36449030 on the Anxa5 transcript exhibits a range of 0–36% editing in the 24 cells (26% in the bulk). While these examples are suggestive of editing heterogeneity, given the stochasticity inherent in single cell data, a more sophisticated statistical framework is required to assess if a specific site is indeed being edited at variable rates.

**Hierarchical Bayesian model for editing rates.** The biological problem to be modelled can be described in the following statistical context (discussed in complete detail in the Supplementary Methods). First, we select an editable site of interest on a given RNA transcript, and its associated probability of editing. The transcript can be modelled as a coin, with a probability of falling on 'Heads' when tossed equal to the associated editing probability (not necessarily 50%). We can think of each single cell $j$ as consisting of a stack of coins (with each coin corresponding to the same site on a different copy of the transcript), each having the same probability of falling on 'Heads', denoted by $p_j$. Finally, we can think of the population of cells as a bag full of stacks of coins,

where the probability of 'Heads' for each stack is drawn from an unknown distribution on [0, 1]. The variance of this distribution quantifies the diversity among editing rates of different cells (or stacks of coins). The single-cell experiments correspond to randomly picking $J$ stacks from the bag and tossing some of their coins, whereas the bulk experiment corresponds to randomly picking a large number of individual coins from the bag, after emptying all the stacks in and mixing them together.

More formally, for a fixed genomic coordinate and given single cell $j$, we denote by $x_j$ the number of edited reads, $n_j$ the corresponding total number of mapped reads and $p_j$ the probability of editing (or editing rate). Similarly, we denote by $x$, $n$ and $p$ the number of edited reads, total number of reads and editing rate, respectively, for the bulk RNA-seq experiment. $v$ will be the variance of editing rate among cells.

The first component of our hierarchical model is a binomial distribution:

$$x_j | p_j, n_j \sim \text{Bin}(n_j, p_j), \; j = 1, \dots, J \quad (1)$$

Using our analogy, this models the probability of observing $x_j$ 'Heads' when tossing $n_j$ coins, where $pj$ is the probability of getting 'Heads.' In this case, the binomial distribution is a natural choice. The second component of the hierarchy is used to couple together the different editing rates $\{p_j\}_{j=1}^{J}$, through

$$p_j | p, v \sim \text{Beta}^*(p, v) \quad (2)$$

Here Beta* represents the beta distribution in the non-standard mean/variance parametrization, as detailed in the Supplementary Methods.

Assuming that the $J$ single cells are randomly sampled from the bulk RNA-seq experiment and that the number of cells $n$ comprising the population of a 'bulk' RNA-seq data set is large (at least $10^5$ cells), the mean editing rate is well approximated by its standard mean estimator

$$p \equiv \frac{x}{n} \qquad (3)$$

That is, since the number of cells $n$ that comprise the population of a 'bulk' RNA-seq data set is typically large, the mean editing rate is accurately approximated by its standard mean estimator.

Note that $v \in [0, \ p(1-p)]$ is highly interpretable, with $v = 0$ corresponding to identical editing rates across cells (no variability) and $p(1-p)$ corresponding to the largest possible editing rate variability allowed by the model.

**Bayesian inference**. The previous section specified a hierarchical statistical model such that, for each edited site, the model is parametrized by the editing rates $\{p_j\}_{j=1}^J$ and their variance $v$. Because we assume that the edited sites can be treated independently, we can restrict attention to just a single genomic location, for which we can estimate the posterior distribution by applying Bayes' rule and the appropriate Markov chain Monte Carlo algorithms (Supplementary Methods).

An important part in building a successful Bayesian model is the choice of the prior. Formally, the prior distribution encodes our *a priori* uncertainty with respect to the model parameters. To derive a prior distribution on the variance among editing rates, denoted by $P(v)$, we take a Bayesian approach to construct 'penalized complexity' (PC) priors, as proposed recently in[18]. PC priors aim to provide useful statistical regularization in the absence of detailed subjective prior knowledge through a principled approach to prior construction. They are predicated on Occam's principle that the simplest 'base' model $\mathcal{B}(0)$ should be preferred to more complex alternatives $\mathcal{B}(v)$. PC priors are therefore well suited to investigate the scientific null hypothesis ($v = 0$).

More specifically, the PC prior is defined by placing an exponential distribution $P(d) = \lambda e^{-\lambda d}$ on the distance $d(v) = \sqrt{2\mathrm{KL}(\mathcal{B}(v) \ || \ \mathcal{B}(0))}$, where $\mathrm{KL}(\mathcal{B}(v)||\mathcal{B}(0))$ is the Kullback–Leibler divergence between the distributions $\mathcal{B}(v)$ and $\mathcal{B}(0)$. This explicitly penalizes model complexity, quantified in terms of distance to the base model. Since the Kullback–Leibler distance is parametrization invariant, it follows that so is the PC prior $P(v)$. The prior specification is completed by assuming that the probability of the variance being greater than some value $v_L$ is smaller than a chosen threshold—in this case, 0.01. In mathematical notation, $P[v > v_L] < 0.01$ (Supplementary Methods and Supplementary Fig. 2). Results in this paper were based on a conservative prior that assumes little variability ($v_L = 0.142$).

The posterior density of $v$ provides a quantification of RNA editing variance. Statistically, this can be formalized through the calculation of highest posterior density (HPD) credible intervals (a Bayesian analogue of confidence intervals), where a 95% HPD interval $C$ for $v$ means that the probability of $v$ falling in $C$ given the observed data $D$ is at least 95%. Thus, HPD intervals enable direct probability statements about the chance of $v$ falling in $C$.

**Sensitivity analysis and validation of Bayesian model**. To test the model, we performed simulations using artificial data sets comprising of $J = 20$ cells exhibiting high levels of editing variance for a theoretical site, generated by randomly sampling the number of edited reads to attain effective editing rates only in the region of $0 - 5\%$ and $95 - 100\%$.

To assess the effect of the number of reads mapping to an edited site on inference, simulations were run for varying levels of coverage while fixing all other parameters (Fig. 2). The first column of Fig. 2 compares the posterior and prior editing rate variances. Even with as few as 10 mapped reads per cell, the model is still capable of detecting the high variance in the data since the posterior mass is shifted away from the vicinity of zero. Similarly, the modes of the posterior editing rates per cell (second column of Fig. 2) exhibit large variation irrespectively of coverage. The third column of Fig. 2, illustrating the posterior and prior marginal editing rates, shows the distribution of editing rates among cells. While for coverage $\geq 20$ we observe clustering of the editing rates in the regions of $0 - 5\%$ and $95 - 100\%$ as expected, this is not so clear for coverage $\leq 10$, indicating that this coverage might not be sufficient to accurately capture variability.

Figure 3 demonstrates in an analogous manner the role of the number of cells in inference. For artificial data with high editing rate variability and with presumed coverage of 20 reads per site per cell in each data set, our ability to learn from data via the model is not perturbed by the number of cells, with even as few as $J = 5$ cells appearing to be sufficient to detect high editing rate variance.

To further validate our model, we ran simulations on two distinct data sets: one with very high variance (Fig. 4a), sampling editing rates in the region of $0 - 5\%$ and $95 - 100\%$, and one with very low variance (Fig. 4b), with editing rates sampled from the region of $(45, 55\%)$. In the former case, the posterior distribution of the variance parameter $v$ (hereafter referred to as simply the posterior variance) is shifted away from zero, as expected for high variance in editing rates, whereas in the latter scenario, it shifts in opposite direction accumulating towards zero. When the variance is high, the posterior editing rates per cell are dispersed. In contrast, they concentrate around the bulk mean of 0.5 when the variance is low. The marginal editing rate histogram exhibits fat tails when the sampled editing rates come from extreme quintiles, while it is peaked around the bulk mean of 0.5 when there is nearly no editing rate variance.

We also ran simulations on the model using editing rates more in the range of what is observed physiologically (Supplementary Fig. 3), using $J = 20$ cells and coverage at 20 reads per site. Minimal variance (mass still centred at 0) is observed when sampling single-cell editing rates over the range of 0–20% (with bulk mean at 10%; Supplementary Fig. 3a) or from the wider range of 20–50% (with bulk mean at 35%; Supplementary Fig. 3b). The posterior variance shifts from 0 when the artificial data set samples from the region of 5–15% and 40–50%, with the shift being more marked when the bulk mean is in between these two regions (bulk mean at 35%; Supplementary Fig. 3c) instead of within one of those ranges (bulk mean at 10%; Supplementary Fig. 3d). This indicates that the bulk mean does have a significant effect on how much variance is observed, which is consistent with expectations: small variability means that editing rates among cells are very similar and consequently also very similar to their average.

In an attempt to test the model on an experimental condition where there should be no editing rate variance, we looked for a heterozygous genomic SNP in the RNA-seq data pooled from the 24 single macrophages sequenced (since the cells were all taken from the same mouse). We screened the pooled RNA-seq data for genomic common, coding biallelic variants in dbSNP 138, and found that the mouse sequenced was potentially heterozygous at 47 sites (that is, expressed both the annotated reference and alternate bases, with the alternate base occurring $15 - 85\%$ of the time). However, coverage across the 24 cells for these 47 sites was poor and the cells exhibited expression from only one of the two alleles for many of the sites, consistent with previous reports[19,20]. Therefore, variance was actually not low

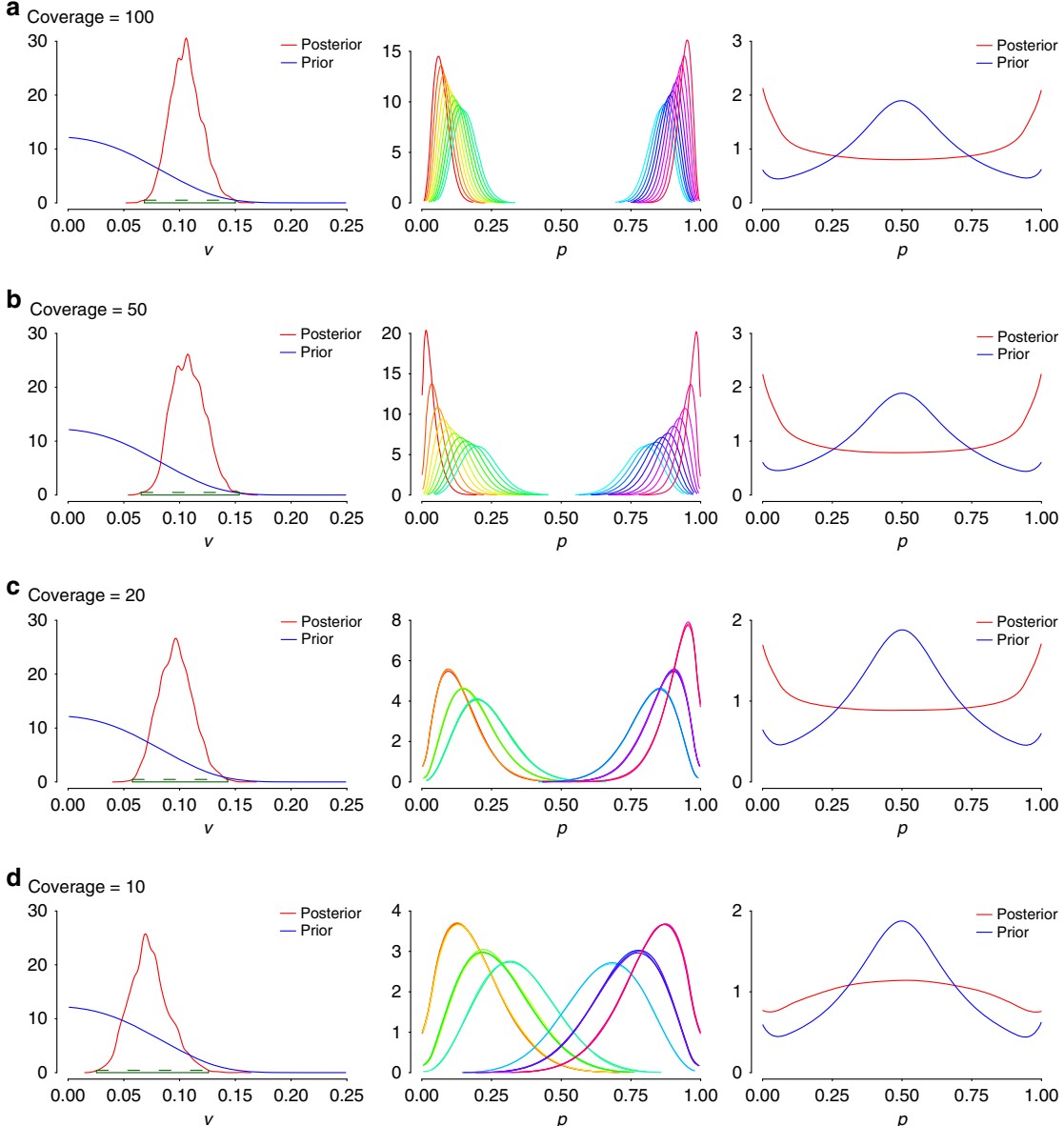

**Figure 2 | Effect of changes in coverage on model.** Histograms of variance of editing rates (left), editing rates (middle) and marginal editing rates (right), for varying coverage ((**a**) 100; (**b**) 50; (**c**) 20; and (**d**) 10 reads per cell, for 20 cells per simulation) using artificial data with high variance.

in this situation as expected, so this did not prove to be a useful control.

**Model suggests editing rates differ in single macrophages**. The use of artificial data allowed us to derive certain constraints: minimally, the model requires some level of local read coverage ($\geq 10$ reads for 20 cells, ideally $\geq 20$ reads) and a minimum number of cells queried ($\geq 5$ cells with a minimum of 20 reads per cell). However, by these standards, most editable sites, identified from bulk RNA-seq data, are insufficiently covered within single-cell data sets to be tested with the model. For example, whereas 275 genes are identified as edited within the bulk macrophage data sets, only 29 of them are consistently detected as expressed in scRNA-seq data (expressed at any level in all 24 single cells), and only 18 sites in 11 genes show sufficient coverage over the editable site by our more stringent guidelines of having at least 20 reads covering a site in at least 20 cells (80 sites exhibit at least 5 cells with at least 20 reads covering a site; and only 9 sites in 4 genes, 6 being in the single transcript B2m, show at least 20 reads in all 24 single cells).

We have chosen to apply the model to three editable sites within three distinct transcripts: B2m (well covered and near 100% edited in bulk RNA-seq data, though the editing rate is 13–40% at six specific positions within the transcript); Anxa5 (well covered, 26% edited in bulk RNA-seq data); and Cybb (well covered, bulk editing rate of 23%). For these sites, we computed the prior and posterior distribution of the variance (plotted in Fig. 5). Further, for these sites we computed the 95% HPD credible intervals, from the posterior distribution of the variance. The computation shows that $v = 0$ is not contained in the 95% HPD interval of the posterior distribution of $v$ for any of the three tested sites. This can be seen as analogous to rejecting the hypothesis 'there is no variability' with $P$ value that is $\leq 0.05$, and is exactly determined by the area under each curve (that is, 95% of the area under the posterior distribution of $v$ is contained within the 95% HPD interval). The non-zero variance indicates that there is heterogeneity in editing rates amongst the 24 cells profiled (Fig. 5).

Note that the Bayesian approach is not limited to testing the hypothesis of $v = 0$. The posterior distribution of the editing rates,

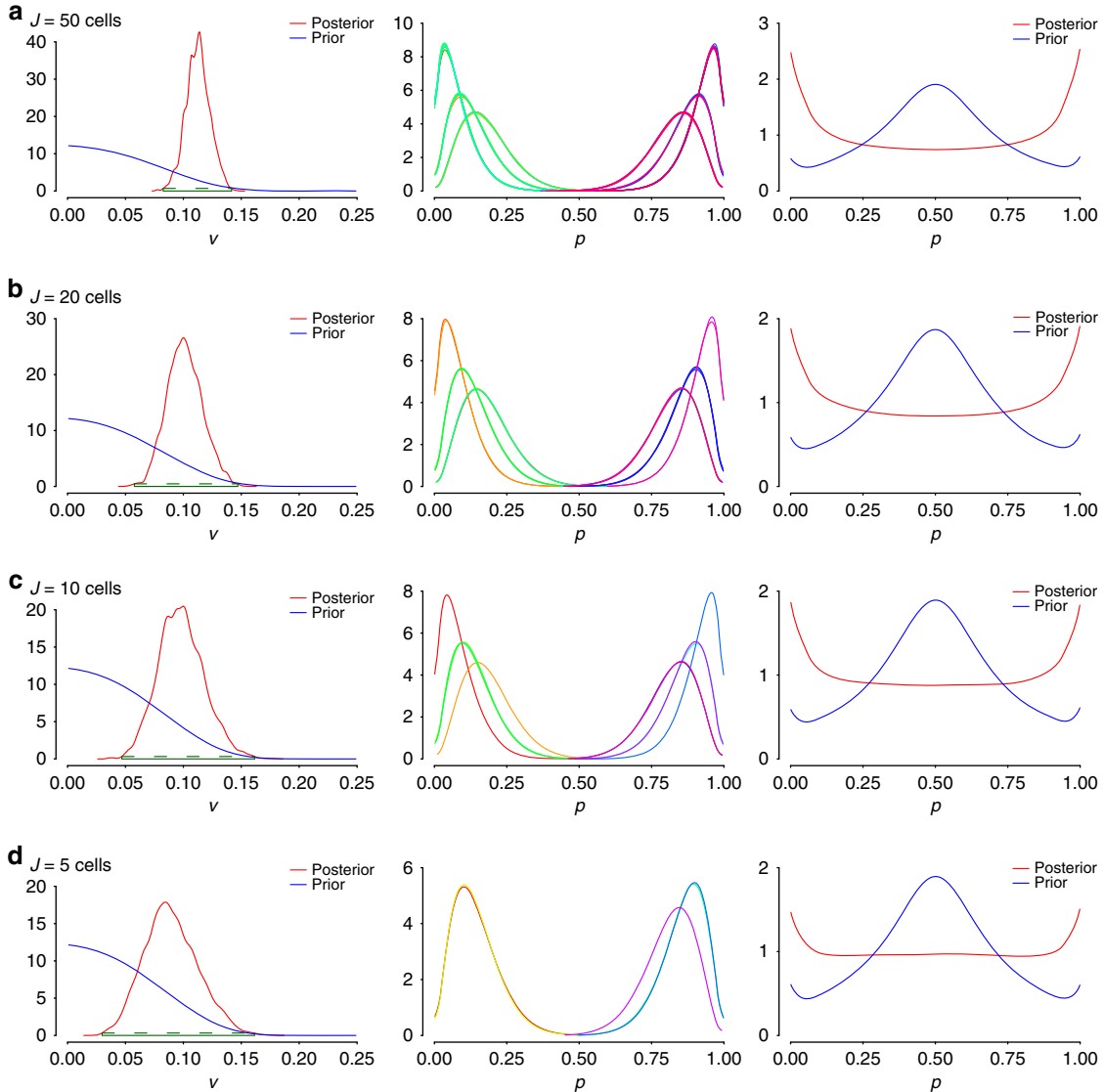

**Figure 3 | Effect of changes in cell number on model.** Histograms of variance of editing rates (left), editing rates (middle) and marginal editing rates (right) for varying number $J$ ((**a**) 50; (**b**) 20; (**c**) 10; and (**d**) 5 cells per simulation, with 20 reads per cell) using artificial data with high variance.

shown in the third column of Fig. 5 is highly interpretable: it can be thought of as a histogram of editing rates of different cells from the bulk experiment. In the case of B2m, the posterior accumulates mostly between [0.25, 0.75], with less mass in the end points, indicating that an arbitrary single cell is likely to be edited with an editing rate ~50%. For Anxa5, we see that most of the mass is concentrated in the [0, 0.5] interval, indicating that editing rates vary but are not likely to be very high (more than 50%). Finally, in the case of Cybb, we see that most of the mass is around 0, with a spread of the remaining mass in the rest of the interval, indicating that the majority of cells are unedited, but that the cells that do edit may do so over a wide range of rates.

**Experimental validation of model-predicted heterogeneity.** To validate our model's predictions of substantial cell-to-cell RNA-level sequence variability, we used a modified reverse transcription (RT)–PCR amplification protocol, to amplify regions of specific transcripts (containing editable sites) using barcoded RT primers from single cells (method diagrammed in Supplementary Fig. 4). These amplicons were cloned into bacteria, followed by

standard Sanger colony sequencing. Because RT primers were barcoded, PCR duplicates could be discriminated, and each cDNA sequence illustrated in Fig. 6 represents sequence information originating from a single contiguous transcript segment within a single cell. A number of sequences with distinct barcodes would represent a fraction of transcripts (and their modifications) within one cell, allowing us to probe specific transcripts with substantial depth.

For this validation work, we chose to focus on three regions surrounding editable sites within three macrophage transcripts: B2m; Anxa5; and Cybb. All three transcripts were well covered in RNA-seq data from the single cells (Fig. 6a) but fall into two distinct classes with regard to our model, with the posterior marginal editing rate histograms exhibiting either negative exponential (for example, Cybb) or unimodal (for example, B2m and Anxa5) distributions (Fig. 5, right). For each of the three gene regions studied, we collected and analysed 20 distinct transcripts per cell, from at least 8 single cells, derived from mice distinct from those that gave rise to the bulk RNA-seq and single-cell RNA-seq data. We recovered two types of amplicons: completely unedited amplicons (denoted in grey in Fig. 6b–d);

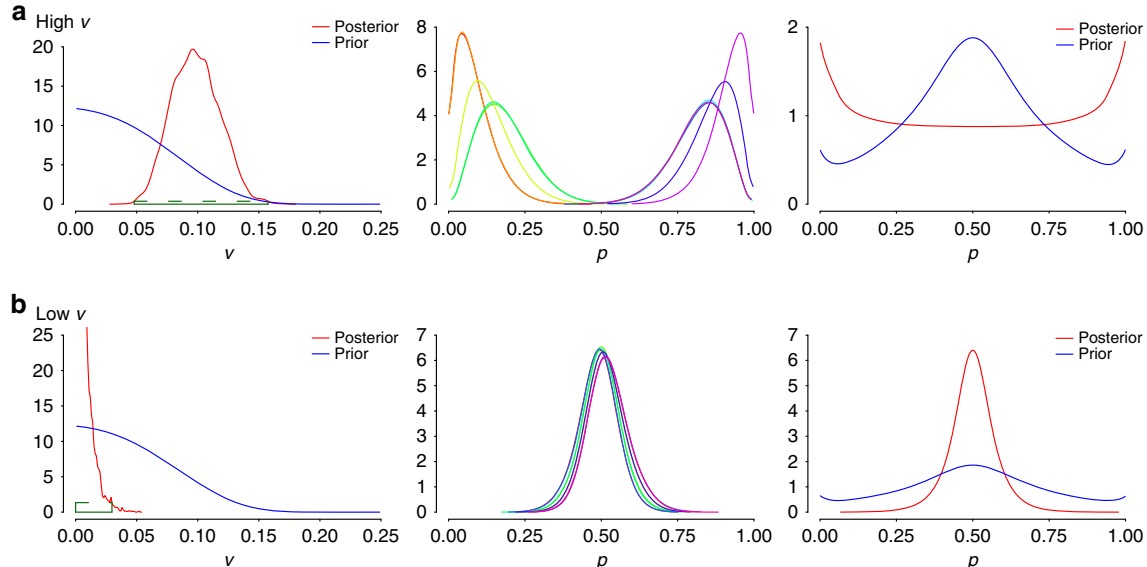

**Figure 4 | Effect of changing the levels of variance on model.** Histograms of variance of editing rates (left), editing rates (middle) and marginal editing rates (right) for two artificial data sets, one with (**a**) high editing rate variance; and the other with (**b**) low editing rate variance.

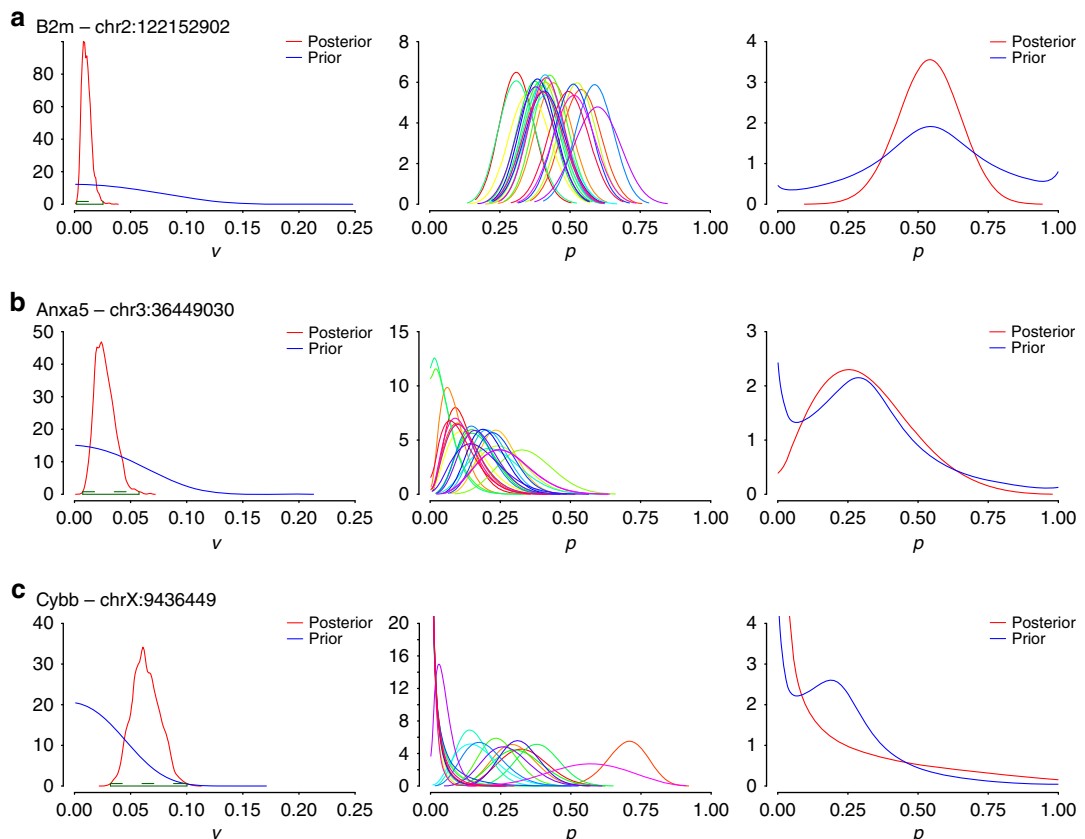

**Figure 5 | Applying the Bayesian model to macrophages.** Simulations run on single-cell data ($n = 24$ data sets) for (**a**) B2m—chr2:122152902, (**b**) Anxa5—chr3:36449030 and (**c**) Cybb—chrX:9436449. Left: histograms of variance of editing rates, across all 24 cells for each site with 95% HDP values, computed from the posterior distribution of the variance, shown as a green dotted line within the plot; middle: histograms of editing rates, denoting the distributions of editing rates for each cell (each cell is labelled with a different colour); right: marginal editing rate histograms, denoting the distribution of editing rates among cells. The posteriors of $v$ and $p$ have been computed using a quadratic interpolation of the respective Kernel density estimators derived from the Gibbs simulations. In particular, the posterior of $p$ has been approximated via Beta kernels, as such a choice naturally adheres to the model assumptions on the distribution of $p$ and further facilitates the estimation of densities with bounded support (see for example refs 37,38).

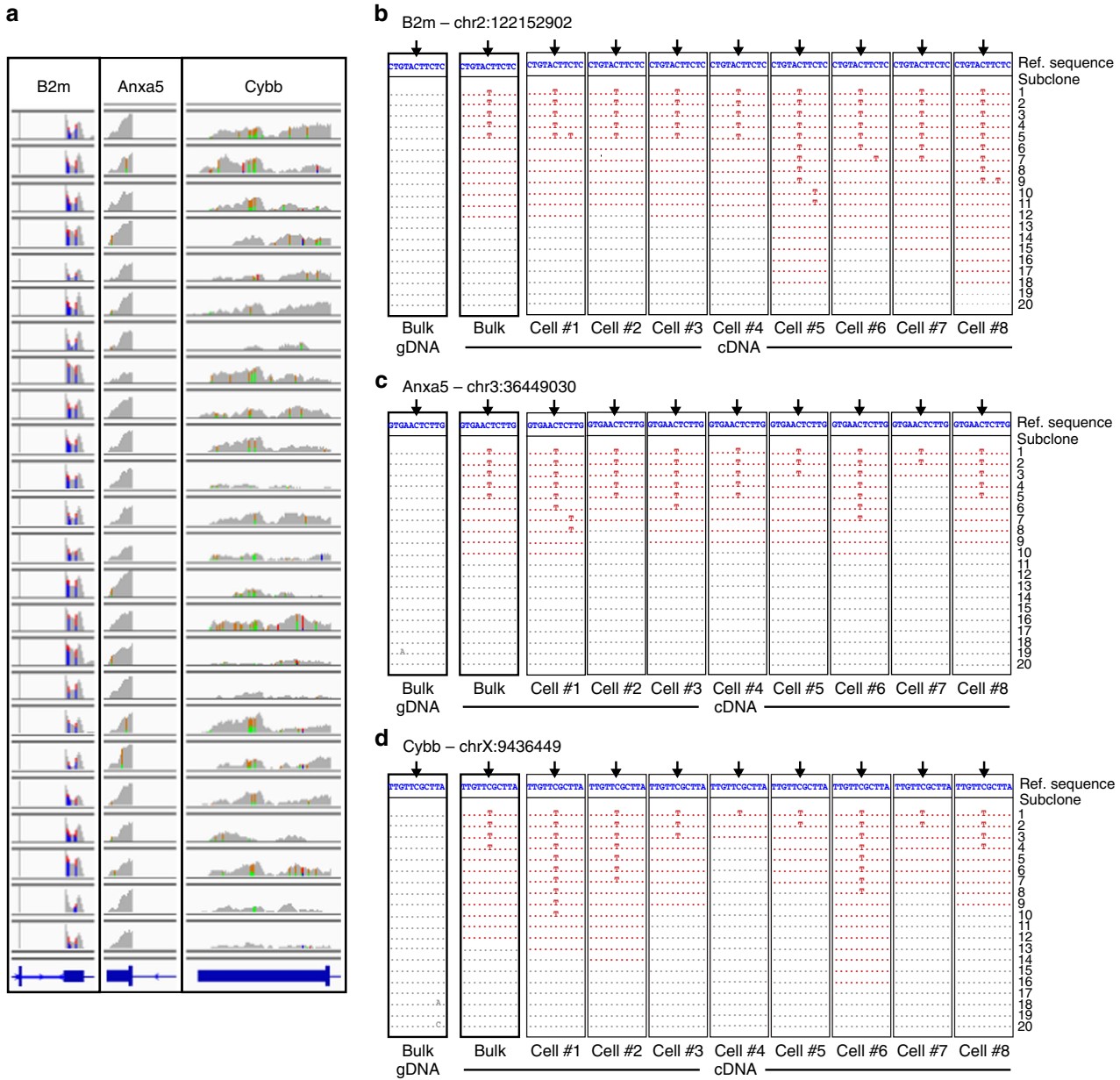

**Figure 6 | Validation of model predictions using targeted amplification of editable sites from single cells.** (**a**) Wiggle plots showing coverage in 3'-untranslated regions for B2m, Anxa5 and Cybb in the 24 bone marrow-derived macrophages profiled. (**b–d**) Sequence alignments from targeted RT–PCR amplification and Sanger sequencing of bacterial colonies for (**b**) B2m, (**c**) Anxa5 and (**d**) Cybb transcripts from gDNA and cDNA from a bulk sample (amplified using standard PCR), and cDNA of single cells (amplified using a modified OneStep RT–PCR protocol, per Supplementary Fig. 4). Alignments, showing the sequence space surrounding a particular editable site, are clustered by sample. Alignments are colour-coded to indicate whether the sequence aligned contained (red) or lacked (grey) editing in the length of the amplicon. Though a C-to-U change may not be shown in the narrow window illustrated, a red sequence would indicate that the amplicon sequence contained at least one C-to-U edit elsewhere (red). Lack of editing in the gDNA indicates that the C-to-U transitions observed are bona fide APOBEC1-mediated RNA editing events.

and edited amplicons (denoted in red, Fig. 6b–d), which in some cases showed editing at multiple, unexpected sites, even when they showed no editing at the site predicted bioinformatically from bulk RNA-seq data (Supplementary Fig. 5a).

Furthermore, the single transcript sequence data we generated with this method align well with model predictions. When we analysed amplicons representing regions in the Cybb transcript, our data demonstrate that cells in which Cybb is nearly fully edited (for example, cell #1, 2 and 6) co-exist in the population with cells that are hardly edited (for example, cell #4). Amplicons for Cybb from additional single cells are shown in Supplementary

Fig. 5b, and indicate a similar distribution. These data are in agreement with the posterior distribution produced by the model (Fig. 5c, column 3). For B2m, although the transcripts are edited in almost every cell, there still is substantial diversity in the location of editing per transcript, and editing at the site queried by the model is ∼50% (Fig. 6b). Similarly, our targeted single-cell amplification and sequencing data indicated that the editable site queried in the Anxa5 transcript was roughly uniformly edited, as predicted by the model. These data also demonstrate that Anxa5 transcripts are substantially hyperedited, which was not apparent from the bulk RNA-seq data. Our data therefore imply that

transcripts that were previously described as edited at a single site from bulk RNA-seq data (due in part to the stringent filtering used in bioinformatics algorithms to identify editing) may actually be edited throughout their length, and could actually be classed as 'hyper-edited.'

**Editing profiles for dendritic cells shift on stimulation.** To investigate if the model could detect the biological regulation of editing rates, we applied our model to a single-cell RNA-seq data set profiling bone marrow-derived dendritic cells under various stimulation conditions[16]. Shalek *et al.* showed that while these dendritic cells are transcriptionally and genomically homogeneous while at rest, they quickly separate into two groups in response to lipopolysaccharide (LPS) stimulation: a group of 'early responders' and a group that eventually catches up later in the response. To investigate if RNA editing may be anticipating the emergent diversity of this population, we applied our model to 20 single-cell RNA-seq data sets and the bulk RNA-seq data set for each of the following conditions: 0 (or unstimulated); 1; 2; 4; and 6 h after LPS exposure[16]. Compared with macrophages, dendritic cells express lower levels of APOBEC1 at the bulk level and are highly variable in terms of APOBEC1 expression at the single-cell level (Supplementary Data 2). We focused on the same sites that were evaluated in macrophages in B2m (Fig. 7), Anxa5 (Fig. 8) and Cybb (Fig. 9), finding that they were generally well covered and edited to some degree in the single dendritic cells as well (Supplementary Data 3). As with the macrophages, we found that there is substantial (or at least, non-zero) variance in editing rates for the selected sites in the 20 dendritic cells assessed at each time point, with the variance interestingly being consistently higher for all three sites in the unstimulated cells compared with cells that had been exposed to LPS for 2 h. Given that the 95% HPD intervals for the posterior distributions of variance for unstimulated compared with 2 h do not overlap for Anxa5 and Cybb suggests that editing at these two sites shows significantly less variability in the stimulated population compared with the unstimulated cells (forest plots in Figs 8 and 9; HPD intervals provided in Supplementary Data 4). The biological relevance of this regulation (and how it occurs) is unclear at the moment, but this result makes some intuitive sense given that unstimulated dendritic cells should be more plastic than cells that have been exposed to a specific pathogen and have already begun going down a specific response pathway. It is also feasible that editing that occurs in a small subset of cells (for instance, Cybb in Fig. 9) at time 0, might 'pre-determine' the fate of those cells (for example, towards becoming early responders) with the rest of the population catching up soon thereafter.

## Discussion

The idea of variability in RNA editing rates as a mechanism for increasing functional heterogeneity across cell populations is an intriguing new possibility that has thus far been largely overlooked. Here, using APOBEC1-mediated RNA editing as a model, we demonstrate that there is a significant range in editing across different single cells, supporting the hypothesis presented in ref. 14. However, transcripts that are edited at a uniform level even within single cells also exist (for example, Anxa5, as predicted by ref. 13).

To arrive at this conclusion, we used RNA-seq to quantify editing in single cells. The normalization of single-cell RNA-seq libraries is the biggest challenge to accurate quantification, given the high variability in capture efficiency from cell to cell[21]. One method to address this problem is to use unique molecular identifiers (UMIs) to tag reads originating from either the 5'- or 3'-end of a given mRNA transcript with a unique sequence[22].

However, since UMIs cannot currently distinguish if reads originating from the interior of a transcript came from different molecules, and C-to-U edited sites are generally covered by internal reads due to RNA editing occurring most frequently at sites distal to the 3'-end, UMIs are not especially useful for evaluating relative rates of editing. Another normalization approach frequently used is to spike-in External RNA Control Consortium (ERCC) standards to control for variation in capture efficiency from library to library[21]. Though useful in comparing the amounts of mRNA captured per cell, ERCC spikes have limited applicability to the problem of editing rates, which relates directly to the amounts of capture of specific transcripts and the sequence variability in between them that is due to editing.

Because of the shortfalls of these experimental approaches, we have used statistical modelling to deconvolute technical and biological effects in assessing the variability of RNA editing rates across single cells. An assumption of the modelling is that RNA-seq samples from the transcriptome in a completely unbiased manner, when in actuality, RNA-seq libraries exhibit a 3'-transcript and GC-content bias[23]. However, it is unlikely that an edited read is more or less likely to be sampled than an unedited read mapping to the same region unless it is substantially hyperedited (or 'ultra' edited[24]), which is often the case with ADAR editing, but not with APOBEC1-mediated editing.

The model is also robust to errors due to RT–PCR, or sequencing. We note that reverse transcriptase has characteristic features (which include error rate—roughly 1/1,500 to 1/30,000 bases[25] and also preferred misincorporation profiles[26,27]). Second, our single-cell Sanger sequencing validation data demonstrate additional non-deaminase-mediated base transitions (for example, Supplementary Fig. 5a), which are likely the result of sequencing error at the level of PCR amplification (estimated to occur at a rate of 1–6 errors per 1,000 bases[28]), and exhibit recurrent C-to-T mismatches from the reference in distinct transcripts, with non-identical barcodes (Supplementary Fig. 5). The likelihood that RT error (and not enzymatic deamination) would give rise to such nucleotide transitions at similar locations within different transcripts is negligible.

In performing the biological validation for this work, we found substantial, largely unexpected editing neighbouring the predicted sites that were simulated. The occurrence of many events over hundreds of base pairs in a single molecule cannot be detected by conventional RNA-seq, which relies on short reads to reconstruct transcripts. The advent of single-molecule sequencing (for example, Pacific Biosciences) should improve our ability to detect editing along specific, contiguous transcripts. Future work will be geared towards modelling editing profiles for all editable transcripts per cell, allowing for a comparison of individual cells by editing signatures, in analogy to population classification based on expression profiles.

Overall, the model presented here can only be used to build reasonable confidence intervals for sites that are highly covered in single-cell RNA-seq data, which limits the number of sites we can evaluate. Poorly covered transcripts are truly representative of few mRNA molecules, exacerbating issues of sampling, especially in the context of editing (Supplementary Fig. 6 and Supplementary Table 2). Thus, our analysis must be strictly limited to well-covered transcripts. Within those constraints, the model predicts that some sites show high variance and we provide experimental validation for one of these (Cybb) here. The finding of one such site (out of only three evaluated) allows us to predict that a subset of transcripts that are edited follow the Pullirch and Jantsch models, and thus that RNA editing does generate sequence heterogeneity within populations. We expect that the technical limitations that result in poor libraries will be substantially ameliorated in the future, rendering this model increasingly

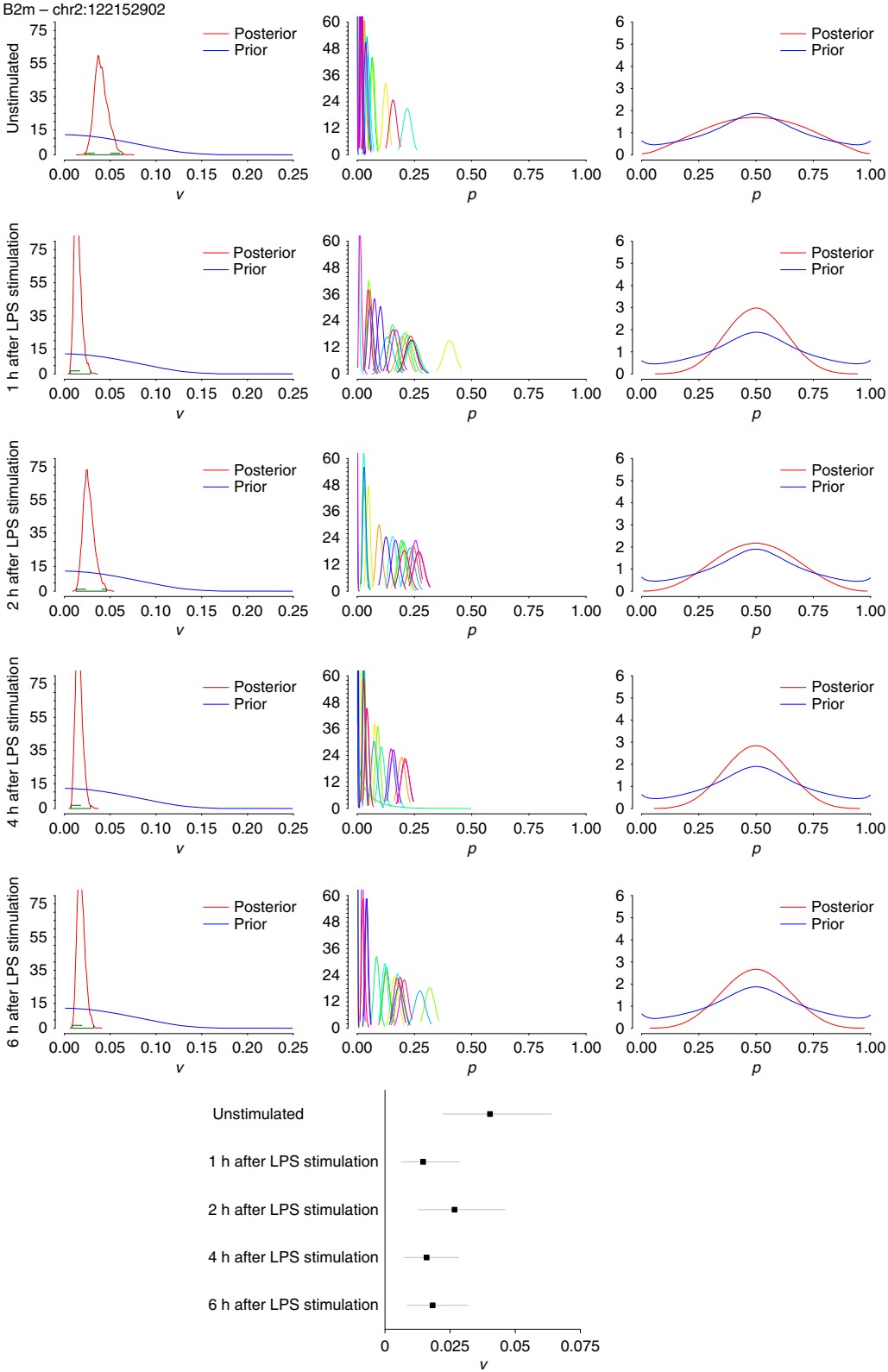

**Figure 7 | Applying the Bayesian model to dendritic cells 0 to 6h after LPS stimulation for edited site in B2m.** Simulations were run on single-cell data ($n = 20$ data sets each) and bulk data (from 10,000 cells) for B2m—chr2:122152902. Left: histograms of variance of editing rates, across all 20 cells for each site with 95% HDP values, computed from the posterior distribution of the variance, shown in green over each plot; middle: histograms of editing rates, denoting the distributions of editing rates for each cell (each cell is labelled with a different colour); right: marginal editing rate histograms, denoting the distribution of editing rates among cells. The posteriors of $v$ and $p$ have been com'puted as described in Fig. 5. Each row represents data sets of 20 cells each under the indicated times from LPS stimulation (all data from ref. 16). A forest plot summarizing the 95% HPD intervals calculated from the posterior distribution of variance is shown in the bottom centre, with the line segment for each condition spanning the lower to upper bound of the HPD interval, and the box indicating the mean variance.

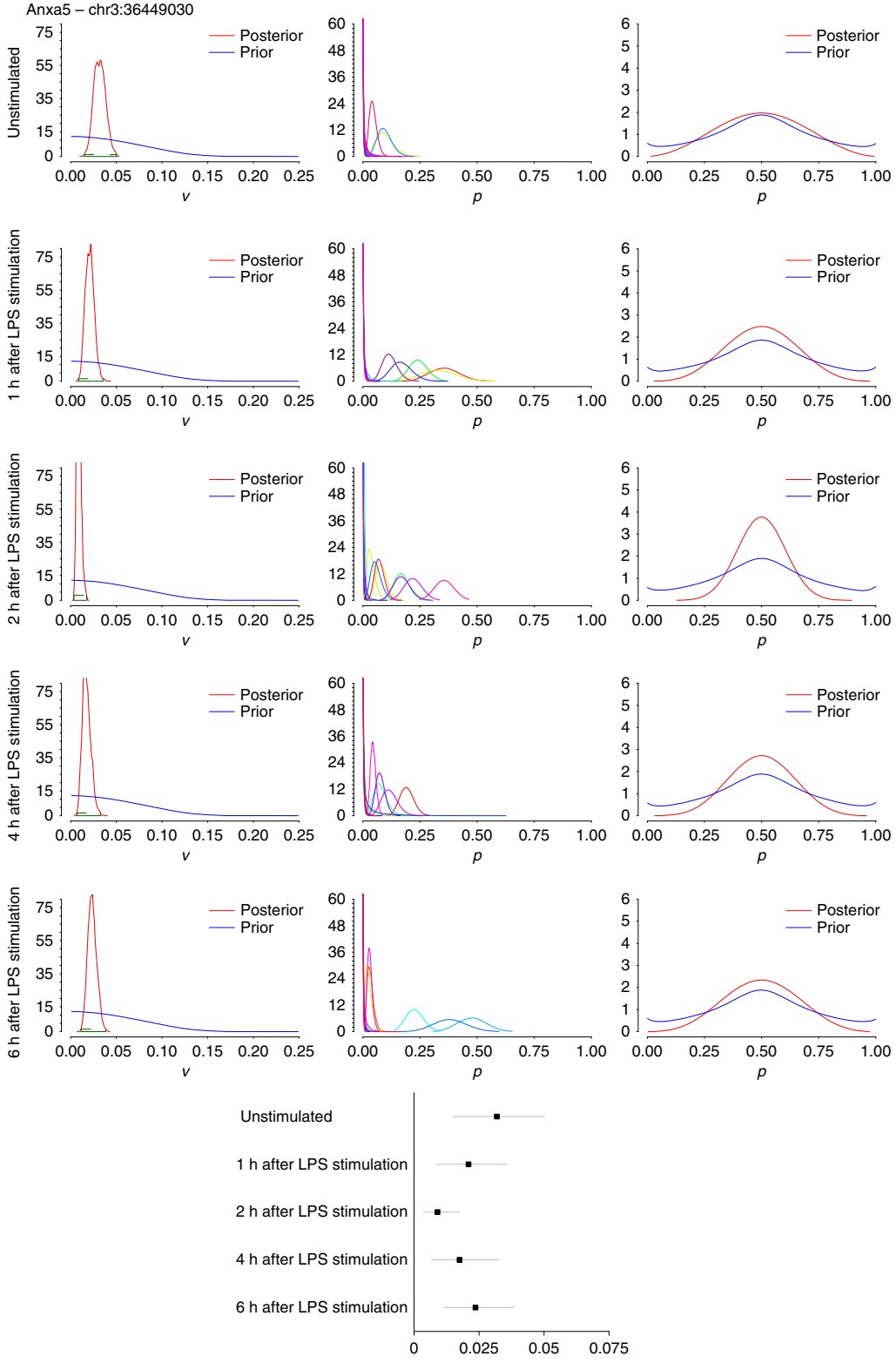

**Figure 8 | Applying the Bayesian model to dendritic cells 0 to 6h after LPS stimulation for edited site in Anxa5.** Simulations were run on single-cell data ($n = 20$ data sets each) and bulk data (from 10,000 cells) for Anxa5—chr3:36449030. Left: histograms of variance of editing rates, across all 20 cells for each site with 95% HDP values, computed from the posterior distribution of the variance, shown in green over each plot; middle: histograms of editing rates, denoting the distributions of editing rates for each cell (each cell is labelled with a different colour); right: marginal editing rate histograms, denoting the distribution of editing rates among cells. The posteriors of $v$ and $p$ have been computed as described in Fig. 5. Each row represents data sets of 20 cells each under the indicated times from LPS stimulation (all data from ref. 16). A forest plot summarizing the 95% HPD intervals calculated from the posterior distribution of variance is shown in the bottom centre, with the line segment for each condition spanning the lower to upper bound of the HPD interval, and the box indicating the mean variance.

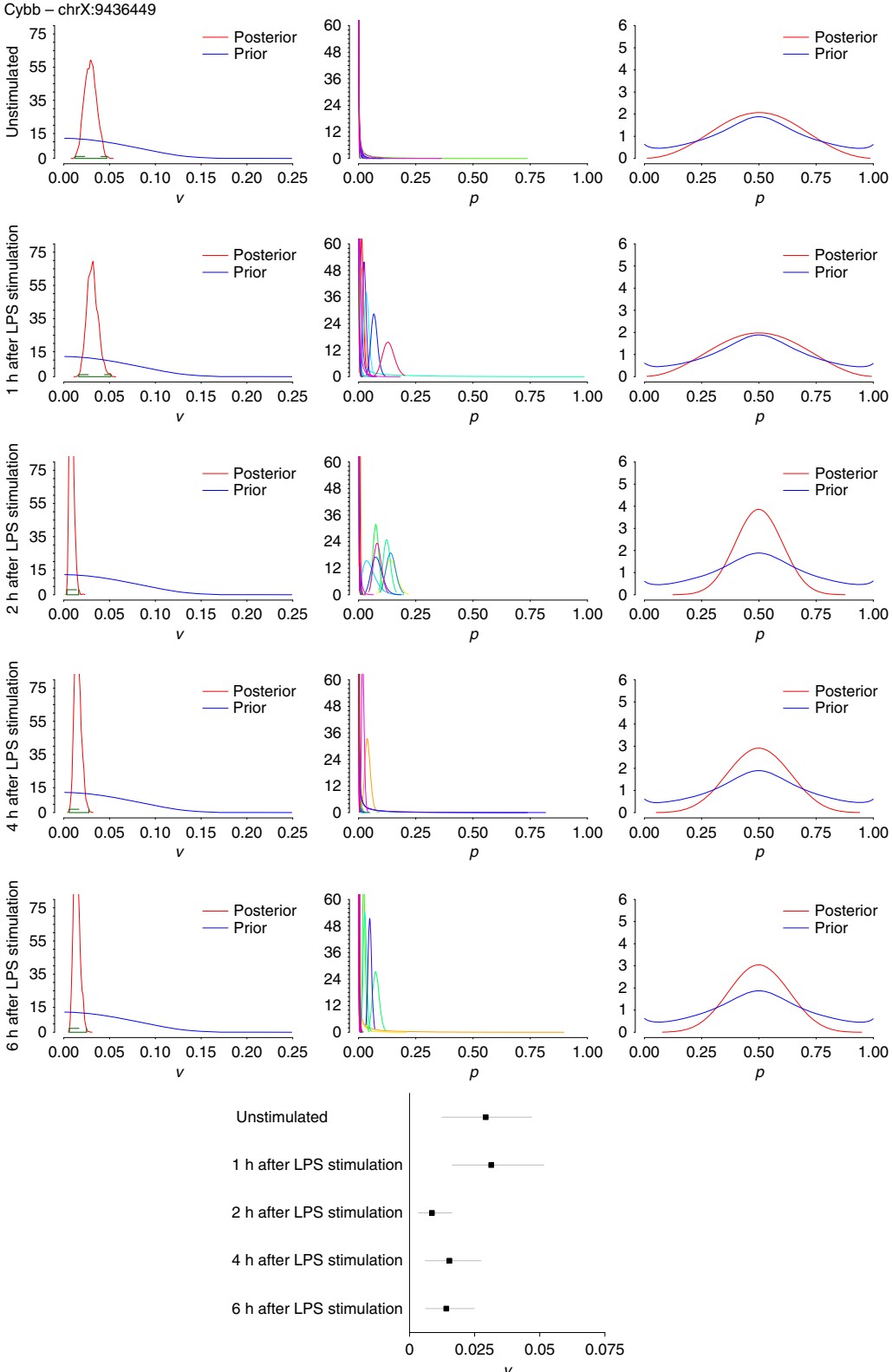

**Figure 9 | Applying the Bayesian model to dendritic cells 0 to 6h after LPS stimulation for edited site in Cybb.** Simulations were run on single-cell data ($n = 20$ data sets each) and bulk data (from 10,000 cells) for Cybb—chrX:9436449. Left: histograms of variance of editing rates, across all 20 cells for each site with 95% HDP values, computed from the posterior distribution of the variance, shown in green over each plot; middle: histograms of editing rates, denoting the distributions of editing rates for each cell (each cell is labelled with a different colour); right: marginal editing rate histograms, denoting the distribution of editing rates among cells. The posteriors of $v$ and $p$ have been computed as described in Fig. 5. Each row represents data sets of 20 cells each under the indicated times from LPS stimulation (all data from ref. 16). A forest plot summarizing the 95% HPD intervals calculated from the posterior distribution of variance is shown in the bottom centre, with the line segment for each condition spanning the lower to upper bound of the HPD interval, and the box indicating the mean variance.

useful. On the basis of our current data, we establish that to accurately capture the full range of editing variability, individual sites have to be covered by at least 20 reads per cell (as a conservative estimate), although as few as 10 reads per cell appears to be sufficient for detecting significant variance in editing rates. We therefore argue that the $50,000 - 100,000$ reads deemed the minimum needed for single-cell differential gene expression[29] are insufficient for the purposes of evaluating the status of epitranscriptomic changes, as they require higher per-base coverage (in contrast to per-gene coverage).

Despite these caveats, we were able to validate experimentally using targeted single-cell RT–PCR the statistical model's prediction of substantial sequence heterogeneity from cell to cell for specific edited sites in macrophages and dendritic cells. This result is especially notable because data from either cell type suggest that there are no significant gene expression differences between individual cells while at rest. Furthermore, we find few if any gene expression differences that are dependent on editing at the population level. However, macrophages and dendritic cells are known to acquire substantially different functional attributes within minutes of stimulation[16]. We hypothesize that the editing-dependent variability in mRNA sequence, which results in robust differences in translation (Rayon-Estrada et al., submitted), anticipates the phenotypic heterogeneity observed on stimulation.

The statistical framework introduced here has numerous potential applications. The model can be used to determine if the editing profiles of specific sites are regulated (as demonstrated in this study for LPS stimulation), for instance, under different environmental conditions, or between developmental or cell cycle states. This model can also be easily adapted to examine other RNA modifications at single-nucleotide resolution, including ADAR-mediated RNA editing, pseudouridylation and m6A methylation. It may be especially useful in the study of single cells taken from tumours (for example ref. 30), which, like immune cells, are known to be highly heterogeneous in terms of function. Indeed, multiple recent papers have shown that RNA editing is elevated in many different types of cancer, and thus may play a role in diversifying the transcriptome to a tumorigenic effect[8–10]. Overall, editing-induced sequence variability at the transcriptome level could prove to be as informative as DNA somatic mutations for cellular differentiation and manifestations of disease.

## Methods

**Data sets.** We prepared RNA-seq libraries from bone marrow-derived macrophages (both bulk and single cell). Single-cell and bulk libraries prepared from dendritic cells that were unstimulated. Dendritic cells exposed to LPS for 0, 1, 2, 4 or 6 h from ref. 16 were also used (data downloaded from NCBI GEO accession number GSE48968—see Supplementary Data 5 for SRA coordinates).

**Cell culture.** Bone marrow was flushed from femurs of 6- to 8-week-old wild-type (Jackson Labs, Bar Harbor, ME) or $APOBEC1^{-/-}$ (courtesy of NO Davidson, Washington University in St. Louis) C57BL/6J mice. To obtain bone marrow-derived macrophages, cells were cultured in DMEM, supplemented with FBS, non-essential amino acids, beta-mercaptoethanol and $20\,ng\,ml^{-1}$ macrophage colony-stimulating factor (M-CSF) (Peprotech, Rocky Hill, NJ) on bacterial plates. One day after initial isolation, the cells were replated at 2 million cells per 10 ml media (per 100-mm dish). Half of the media was replaced every 3–4 days. Seven to nine days after initial isolation, the cells were checked for maturation by FACS (Cd11b +, F4/80) and collected.

**RNA-seq library preparation.** To generate single-cell libraries, wild-type bone marrow-derived macrophages were flowed into a C1 IFC for mRNA seq (10–17 μm) chip using the Fluigidm system. Lysis, RT and PCR were performed using the SMARTer Kit designed for the C1 (Clontech, Mountain View, CA). The efficiency of chip loading (capture) was confirmed by microscopy and any of the wells that contained either none or more than one cell were noted and discarded from further library preparation and analysis. In all, 24 single cDNA libraries were selected for library preparation on the basis of concentration and size range, as

determined via Agilent Bioanalyzer. Sequencing libraries were made from the cDNA using the Nextera XT DNA Sample Preparation Kit (Illumina). To generate conventional bulk RNA-seq libraries, total RNA was extracted using Trizol (Invitrogen, Grand Island, NY) from 500,000 to 1,000,000 macrophages of each genotype (wild type and $APOBEC1^{-/-}$). A unit of 1 μg of the total RNA collected per condition was then treated with DNase, and then processed using the NEBNext Ultra Directional RNA Library Prep Kit for Illumina (NEB, Ipswich, MA).

**Sequencing and alignment.** Libraries were sequenced on the Illumina HiSeq 2000, generating 100-nucleotide, single-end reads. Reads were trimmed for quality and adapters using Trim Galore!, and then aligned to mm10 (using the reference sequences and annotations provided by iGenomes (Illumina)) using Tophat2 (ref. 31), allowing only unique alignments and up to 2 mismatches per 20-nucleotide segment. Gene expression was determined using Cufflinks[32]. The correlation of gene expression between the ensemble of single cells and the bulk was calculated using the Spearman and Pearson correlation coefficients. Potential PCR duplicates were removed from single cell alignments using Picard (http://broadinstitute.github.io/picard).

**RNA editing detection pipeline.** APOBEC1-dependent C-to-U RNA editing was detected from the bulk RNA-seq data sets using a modified version of our previously published bioinformatics pipeline[3] (Supplementary Fig. 1 and Supplementary Table 1). Briefly, a vector consisting of the number of A's, T's, G's and C's that occurred at each coordinate was constructed from the SAMtools pileup[33] for both the wild-type and $APOBEC1^{-/-}$ bulk RNA-seq alignments that were deduplicated by Picard (http://broadinstitute.github.io/picard). For each coordinate that exhibited a C-to-T change only in the wild-type sample (that is, not in the knockout alignment, such that only APOBEC1-dependent C-to-T changes are kept) and met a number of stringent quality control thresholds (minimum of five reads covering site, with at least two reads supporting the editing event, excluding sites that showed multiple types of transitions; and discarding reads that contain indels, support an edit in the first or last two base pairs of a read), the angle between the corresponding vectors for the wild-type and knockout were compared. Putative hits were retained if the magnitude of the wild-type vector was at least 15 and the angle between the wild-type and knockout vectors was at least 0.11 radians (approximately equivalent to a minimum coverage of 20 reads and an editing rate of 10%). Potential sites were also filtered against genomic DNA-derived SNPs in dbSNP138, and removed if they occurred within four base pairs of a splice junction (using the exon junctions compiled by the Zhang lab, using OLego[34]) or in simple or tandem repeats (softmasked regions by RepeatMasker). Reads supporting edits were run through BLAT[35] to ensure that they were not ambiguously mapped. The pipeline was programmed using Bash and Python with the Pysam (https://github.com/pysam-developers/pysam) pileup engine. RNA editing events were then validated by designing primers proximal to the sites of interest, amplifying those regions from cDNA and genomic DNA from both wild-type and $APOBEC1^{-/-}$ cells, and performing colony sequencing.

**Targeted single-cell RT–PCR.** To analyse editing of specific sites in single cells, single wild-type macrophages were sorted into 96-well PCR plates with 5-μl of lysis buffer, containing 0.45% NP-40, $0.36\,U\,\mu l^{-1}$ RNAse Inhibitor and $0.18\,U\,\mu l^{-1}$ Superase-In (Ambion). RT–PCR amplification was done with gene specific primers and the OneStep RT–PCR kit (Qiagen), using a modified protocol. Single-transcript molecules were tagged with barcoded gene-specific primers that have an additional universal sequence, used in RT. These primers were then digested with 1 U of Exonuclease I (30 ºC for 30 min; NEB)[36]. Afterwards, a mix of universal forward and gene-specific reverse primers were added to the PCR mix and 35–40 cycles of PCR were performed. The PCR products were introduced into bacteria using a TOPO TA cloning kit (Invitrogen) and single bacterial colonies were sequenced using Sanger sequencing. The resulting sequences were then aligned to the reference transcriptome (Macvector) and PCR duplicates were eliminated using the barcodes.

Transcript specific primer sequences were as follows: B2m primers: RT: 5′-GGCCAGTGAATTGTAATACGACTCACTATAGGNNNNNNNAAAGCAG AAGTAGCCACAGGGTTG-3′. PCR-reverse: 5′-TTAAGCATGCCAGTATGGC CGA-3′; Anxa5 primers: RT: 5′-GGCCAGTGAATTGTAATACGACTCACTA TAGGNNNNNNNGTCGCCAATGTTTTGGAT ACTACCATC-3′. PCR-reverse: 5′-GCGACACATCTGGAGACTATAA GAAGGC-3′; Cybb primers: RT: 5′-GGC CAGTGAATTGTAATACGACTCACTATAGGNNN NNNGAGGGTTTGTG CCTAGTCTTATTGCA-3′. PCR-reverse: 5′-GCATGCGCTCATCTTGTTTT GACTTC-3′. Universal PCR-forward: 5′-GGCCAGTGAATTGTAATACGACTC ACTATAGG-3′.

**Statistical methods.** For full methods, please see Supplementary Methods. Artificial data were generated by randomly sampling the number of edited reads to attain effective editing rates in the desired ranges (for example, for cell $j$, 20 of 20 reads may be randomly selected as edited, for 100% editing within that cell). An implementation of our model is publicly available as a package written using the Julia programming language: https://github.com/scidom/CellwiseEditingDifferentiation.jl

**Data availability.** Bulk and single-cell macrophage RNA-sequencing data are publicly available in the NCBI GEO repository under the accession number GSE74720. The experimental data used to generate the graphs presented in the paper are provided in Supplementary Data 1 (macrophages) and Supplementary Data 3 (dendritic cells).

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

## Acknowledgements

D.H. was supported by NIH F32CA183318. T.P. and A.P. were supported by Leverhulme project grant RPG-2013-270. V.R.E. and F.N.P. were supported by Starr Cancer Consortium grant I7-A767. C.J.O. was supported by UK EPSRC grant 'Centre for Research in Statistical Methodology' EP/D002060/1. T.P. and C.J.O. performed this work while they were at the University of Warwick. We gratefully acknowledge Xiaofei Ye for contributions to developing the barcoded single-cell RT–PCR protocol used here (Supplementary Fig. 4).

## Author contributions

D.H. and F.N.P. formulated the biological problem; T.P., C.J.O. and A.P. formulated the statistical problem and developed the model; T.P. implemented the model and performed the sensitivity analysis and validation; D.H. prepared bulk RNA-seq libraries from macrophages, performed the bioinformatics analysis and ran the simulations on experimental data; V.R.-E prepared the single-cell RNA-seq libraries from macrophages and validated the model using targeted RT–PCR; D.H. and T.P. wrote the paper, with contributions from all other authors.

## Additional information

**Accession codes:** Bulk and single-cell macrophage RNA sequencing data are publicly available in the NCBI GEO repository under the accession number GSE74720.

**Competing financial interests:** The authors declare no competing financial interests.

