## [Peer Review File · Nature Communications]

Reviewer #1 (Remarks to the Author):

The manuscript by Harjanto et al. explores C to U RNA-editing patterns within several transcriptomes through the analysis of single cell and bulk population RNA-Seq data. The single main conclusion of the manuscript is that there is that there is substantial variation in RNA editing frequencies between single cells. The authors then speculate that this diversity could contribute to some functional heterogeneity of innate immune cells. The authors do not provide any evidence for the functional significance of varied editing rates and thus the manuscript does not go beyond the presentation of the phenomenon. In this respect I feel the manuscript is not suitable for Nature Communications. I am also not sure that analysing 24 cells of a single type using a single library method is sufficient to validate the method and draw such broad conclusions. I also don't understand why an internal or good control for editing rate was not used such as the bone fide physiologically Apolipoprotein B transcript in cells and conditions where it would be highly edited. In my opinion, with more samples of distinct cell types presented, the manuscript would be suited to a more specialized methods journal as the biological meaning of the observations are not tested.

Reviewer #2 (Remarks to the Author):

The authors have used single cell RNA-Seq and targeted single cell qRT-PCR to measure single cell RNA editing rates for specific transcripts. They have performed detailed statistical modeling to provide evidence that individual cells exhibit different RNA editing patterns. Overall, I have four major comments about this manuscript:

1) In my mind, the simplest explanation for the observation that different cells contain RNA molecules that encode the same transcript but with different sequences is that they have different genomes. Assuming these cells have a normal, diploid karyotype, then they can harbor two possible genomic sequences for each gene. In addition, different cells may have different genomes. Furthermore, different cells may have different karyotypes, allowing them to harbor more than two possible genomic sequences for each gene. As far as I can tell, the authors do not control for the possibility that these cells are not genetically homogeneous or the possibility that these cells do not harbor multiple sequences for the same gene on their genomes. The authors present an experiment in which they amplify specific transcripts from individual cells and incorporate unique barcode sequences into their cDNA products so as to distinguish PCR duplicates by sequencing. It seems

that it would be straightforward to use this same approach to verify the gDNA sequences of the corresponding loci.

2) A second possible explanation for the observation that different cells contain RNA molecules that encode the same transcript but with different sequences is that the reverse transcriptase introduces errors in the cDNA during reverse transcriptase. Such an event would be propagated through PCR amplification and sequencing of the resulting library. Reverse transcriptases have notoriously high error rates and typically do not contain proofreading domains to correct misincorporation errors. Is this possibility accounted for by the proposed statistical model? I could not find any mention of this common problem in the paper. It should be noted that, while the barcoding approach used by the authors in their targeted amplification experiments can compensate for PCR duplication, sequencing errors, and PCR errors, it cannot be used to detect errors introduced by reverse transcriptase.

3) One of the conclusions of this manuscript is that different cells exhibit different RNA editing rates. I think that one needs to be more careful with the language used to make this claim. The authors are investigating RNA editing in individual cells at a specific point in time. While there may be evidence of differences in the number of edited transcripts among individual cells at a specific point in time, this does not necessarily mean that different cells edit their RNA at different rates. A cell may contain many edited transcripts at one point in time and then have far fewer edited transcripts at another point in time. In fact, the cells in these populations may, over time, exhibit the same average RNA editing rates.

4) Previous studies have established based on population-level RNA-Seq that RNA editing occurs. As such, I found the title of this manuscript to be inappropriate. It is already well-known that "RNA editing generates sequence diversity within cell populations". It seems that the distinguishing aspect of this work has to do with differences in RNA editing patterns among individual cells.

Reviewer #3 (Remarks to the Author):

This manuscript reports a study of RNA editing variation between single cells. Using single-cell RNA-seq, bulk RNA-seq and statistical modeling, the authors concluded that substantial variability in editing rates exists between single cells. This is an interesting study that addresses a long-lasting question in the field. The problem associated with low read coverage per editing site in single cells is somewhat alleviated by the statistical approach to estimate variation. However, the limitation of the study is that a very small number of editing sites was examined, the results of which may not represent a global picture of editing variability in cell population.

The statistical model is an important component of this work. More detailed and

quantitative evaluation of its performance is needed. For example, how accurate can the model estimate variance v ? In the simulation analysis, does the estimated v correspond to expected v values accurately? If the simulated editing rates are in 0-5% and 95-100%, why is the estimated v much lower than expected? The cases of editing rates being in 0-5% and 95-100% are extreme cases of high variability, what about more realistic cases such as editing rates being 20%-50%? Indeed, different ranges of editing rates should be simulated to evaluate the model performance.

Just for testing purpose, can the authors analyze allelic ratio of heterozygous SNPs (expected to be 1:1) in their data, and determine if variance estimate is small as expected?

There seems to be unexplored values of the modeling approach. It is not surprising that editing rate variability exists across cells, and the level of variability may differ for different editing sites. To strengthen the aspect of biological relevance of this work (which is rather weak in the current version), can the authors extend the models to determine, for a given site, whether it is under certain regulation for low (or high) cell-to-cell variability in editing rate? In other words, is it possible to determine whether the variability of an editing site demonstrated in the data is significantly higher or lower than expected by a non-regulated statistical model? This type of modeling extensions may add significant value to this work.

In the RT-PCR validation, it is intriguing to observe C-to-U changes that are only in single-cells, not in bulk. Are these real C-to-U editing sites? Are they associated with sequence or structural features characteristic of true C-to-U editing? How many other types of nucleotide changes were observed for each experiment? Could these single-cell-specific changes be caused by RT mistakes, or other types of artifacts?

To provide more convincing validation data, the authors should determine whether the variation (and editing rates) observed in RT-PCR faithfully reproduced what they observed in the sequencing data combined with statistical modeling. That is, a more quantitative assessment of the validation data is needed, which may necessitate include of more than 8 cells for each site.

There are quite a number of typos in the text that need to be corrected. In addition, technical parts of the writing are sometimes obscure. For example, what is "p value that is a fraction of 0.05"? Rather than stating it in this way, a specific p value or p value range should be given.

Reviewer #4 (Remarks to the Author):

The manuscript entitled "RNA editing generates sequence diversity within cell populations." by Harjanto et al is dealing with the intriguing issue of exploring RNA editing at the single cell data. The question this paper is dealing with, touches one of the most fundamental aspects of biological diversity, one that created by post transcriptional modifications at the level of single cell. The present picture about these modifications is clearly far from the real biological states and masked by the current methods that averaged millions of cells into one measurement. The recent rapid development of single cell sequencing has the potential to reveal the true nature of post-transcriptional modifications. For example, when we see RNA editing at 20% ratio at a site in a tissue level, with standard RNAseq technique, is it mean that all cells have 20% editing or it means that some cells has 100% at the site and other has 0%?

As far as I know, this paper is the first work that touches this fundamental issue, focusing mainly on C-to-U RNA editing. The paper is interesting, deals with an important subject and the authors clearly present the motivation for the project. I was a referee of the same paper at an earlier version in other journal. At this current version the author address all the concerns I raised there and I believe that this version is much better and I have no additional comments.

Reviewer #1 (Remarks to the Author)

Original concerns not addressed and thus opinion not changed.

Reviewer #2 (Remarks to the Author)

The authors have addressed my comments regarding the possibility of polymorphisms and errors incurred during reverse transcription. As far as the title goes, I would avoid claiming that the population of cells is "otherwise homogeneous" and go with something closer to the first option that the propose in response to comment 4.

Reviewer #3 (Remarks to the Author)

Response to Reviewers' comments:

Reviewer #1 (Remarks to the Author):

The manuscript by Harjanto et al. explores C to U RNA-editing patterns within several transcriptomes through the analysis of single cell and bulk population RNA-Seq data. The single main conclusion of the manuscript is that there is substantial variation in RNA editing frequencies between single cells. The authors then speculate that this diversity could contribute to some functional heterogeneity of innate immune cells.

We thank the reviewer for their comments. Indeed, this particular manuscript is not focused on functional significance (which remains unclear for the vast majority of editing events that we and others have reported). Physiological relevance is the topic of a separate paper that we currently have pending, and would be happy to attach for the reviewer's perusal. The key contribution of the current paper is the development of the statistical tools for quantifying variability of editing rates that allowed us to robustly describe the phenomenon, which is of relevance in its own right. In that regard the reviewer raises two issues (below).

The authors do not provide any evidence for the functional significance of varied editing rates and thus the manuscript does not go beyond the presentation of the phenomenon. In this respect I feel the manuscript is not suitable for Nature Communications. I am also not sure that analysing 24 cells of a single type using a single library method is sufficient to validate the method and draw such broad conclusions.

We note here that the analysis of 24 single cells has launched this investigation – but that the investigation is not limited to 24 cells. Whereas these, together with bulk measurements have set the parameters for the model, validation of the findings was done independently in additional single cells from distinct cultures of macrophages from biological replicate sources (different animals).

I also don't understand why an internal or good control for editing rate was not used such as the bone fide physiologically Apolipoprotein B transcript in cells and conditions where it would be highly edited. In my opinion, with more samples of distinct cell types presented, the manuscript would be suited to a more specialized methods journal as the biological meaning of the observations are not tested.

Regarding controls for editing, the canonical target for C-to-T editing mediated by APOBEC1 (i.e. Apolipoprotein B – originally characterized in the liver and small intestine) is not expressed in macrophages, and so isn't really an appropriate control. We did try using a heterozygous SNP as an internal control, as discussed later in this rebuttal (see response to Reviewer#3).

Reviewer #2 (Remarks to the Author):

The authors have used single cell RNA-Seq and targeted single cell qRT-PCR to measure single cell RNA editing rates for specific transcripts. They have performed detailed statistical modeling to provide evidence that individual cells exhibit different RNA editing patterns. Overall, I have four major comments about this manuscript:

1) In my mind, the simplest explanation for the observation that different cells contain RNA molecules that encode the same transcript but with different sequences is that they have different genomes. Assuming these cells have a normal, diploid karyotype, then they can harbor two possible genomic sequences for each gene. In addition, different cells may have different genomes. Furthermore, different cells may have different karyotypes, allowing them to harbor more than two possible genomic sequences for each gene. As far as I can tell, the authors do not control for the possibility that these cells are not genetically homogeneous or the possibility that these cells do not harbor multiple sequences for the same gene on their genomes. The authors present an experiment in which they amplify specific transcripts from individual cells and incorporate unique barcode sequences into their cDNA products so as to distinguish

PCR duplicates by sequencing. It seems that it would be straightforward to use this same approach to verify the gDNA sequences of the corresponding loci.

We thank the reviewer for this comment, and the clarification that it prompted.

First, the cells we are testing are completely homogeneous (they are non-dividing, primary cells and acquire identical levels of maturation markers before being tested; they are not from cell lines or cancerous cells where comparatively high levels of genetic drift are expected). We have added a cartoon of their derivation as well as their maturation FACS plots (Figure 1a,) noting (in the legend) (a) that they have exited cell cycle (G0) and (b) that they are terminally differentiated (homogeneously F4/80+CD11b+).

In addition, we have amplified genomic DNA (from the population in bulk) with the same primers used for the single cell RT-PCR, to demonstrate that genomic DNA is free of polymorphisms at the location where we detect editing (one example was provided in the previously submitted version of Figure 1 (now in Figure 1b), but we are providing additional examples now in Figure 3). While we cannot test RNA and DNA from the same cell, we do think that the fact that the genomic DNA sequence at these locations matches the reference while RNA does not (at the population level) ameliorate the reviewer's concern.

2) A second possible explanation for the observation that different cells contain RNA molecules that encode the same transcript but with different sequences is that the reverse transcriptase introduces errors in the cDNA during reverse transcriptase. Such an event would be propagated through PCR amplification and sequencing of the resulting library. Reverse transcriptases have notoriously high error rates and typically do not contain proofreading domains to correct misincorporation errors. Is this possibility accounted for by the proposed statistical model? I could not find any mention of this common problem in the paper. It should be noted that, while the barcoding approach used by the authors in their targeted amplification experiments can compensate for PCR duplication, sequencing errors, and PCR errors, it cannot be used to detect errors introduced by reverse transcriptase.

We thank the reviewer for this comment and have added a paragraph in the discussion that addresses this point.

Firstly, we note that reverse transcriptase (RT) has characteristic features (which include error rate - roughly 1/1,500 to 1/30,000 bases (Roberts et al. Science 1988) and also preferred misincorporation profiles (Roberts et al, MCB 1989 and Pathak and Temin, PNAS 1990).

Secondly, our own single cell validation data (Figure 3, Figure S8) demonstrate (a) additional errors, which are likely the result of sequencing error at the level of PCR amplification, as the reviewer notes (this is estimated to occur at a rate of 1 to 6 errors per 1,000 bases (0.1-0.6%) (Wall et al. Genome Research 2014), and is concordant with our data. (b) recurrent mismatches from the reference, that are C-to-T changes and occur at distinct transcripts (with non-identical barcodes). The likelihood that RT error (and not enzymatic deamination) would give rise to such nucleotide transitions at similar locations within different transcripts is negligible.

3) One of the conclusions of this manuscript is that different cells exhibit different RNA editing rates. I think that one needs to be more careful with the language used to make this claim. The authors are investigating RNA editing in individual cells at a specific point in time. While there may be evidence of differences in the number of edited transcripts among individual cells at a specific point in time, this does not necessarily mean that different cells edit their RNA at different rates.

Whereas the reviewer is likely correct in their observation that editing might change with time (due to cell cycle changes, environmental cues, etc. – see supplemental figures S9-S11), the cells that we are sampling here should be as homogeneous as can be (e.g. see response to issue #1, above, now noted in Figure 1a and described within the legend). Thus, in this setting, any differences observed are not due to

the cells being out of sync or otherwise perturbed externally.

A cell may contain many edited transcripts at one point in time and then have far fewer edited transcripts at another point in time. In fact, the cells in these populations may, over time, exhibit the same average RNA editing rates.

While within an individual cell there might be some fluctuation in the fraction of edited transcripts over time (something that is not possible to measure as we cannot sample the same cell twice), the distribution we observe, within an unperturbed population as a whole, does not change with time (i.e. the spread we see is "stable" for the population as a whole, even if individual cell rates change over time). Experimentally, this is indicated by the fact that model predictions from the single cell RNA seq data is validated by Sanger sequencing of PCR amplicons from single cells (Figure 3 and Figure S8) -- even though the single primary cells that were interrogated were derived from different mice and at different times (i.e. the cells were harvested months apart), and sequenced with different methods (next gen sequencing vs Sanger).

4) Previous studies have established based on population-level RNA-Seq that RNA editing occurs. As such, I found the title of this manuscript to be inappropriate. It is already well-known that "RNA editing generates sequence diversity within cell populations". It seems that the distinguishing aspect of this work has to do with differences in RNA editing patterns among individual cells.

We thank the reviewer for this comment, and are happy to revise the title. Three possibilities are listed below:

- a) "RNA editing generates cellular subsets with diverse sequence within populations"
- b) "RNA editing patterns vary within cells of an otherwise homogeneous population"
- c) "Statistical analysis of RNA editing patterns within cells of an otherwise homogeneous population"

Reviewer #3 (Remarks to the Author):

This manuscript reports a study of RNA editing variation between single cells. Using single-cell RNA-seq, bulk RNA-seq and statistical modeling, the authors concluded that substantial variability in editing rates exists between single cells. This is an interesting study that addresses a long-lasting question in the field. The problem associated with low read coverage per editing site in single cells is somewhat alleviated by the statistical approach to estimate variation. However, the limitation of the study is that a very small number of editing sites was examined, the results of which may not represent a global picture of editing variability in cell population.

The reviewer is correct in that a small number of editing sites were examined, because of poor overall coverage of most of the sites, as discussed. As a result, we have definitely not profiled the entire pattern of edits within a single cell. Arguably, even if only a handful of transcripts were differentially edited that should be sufficient as proof of principle of the point we are trying to make (that otherwise homogeneous cells are differentially edited). To address the reviewer's point, we performed Sanger sequencing of molecular-barcoded amplicons from single cells for *Rac1*, a transcript that ought to be differentially edited based on the observed bulk and single cell editing rates as derived from the high throughput sequencing data, but for which coverage in the HTS data did not meet the model's requirements (as determined by sensitivity analysis which indicated that a cut-off of 20 reads per cell was reasonable Figure S2) in a majority of single cells sequenced. While we could generate *Rac1* amplicons from single cells using the method outlined in Figure S7, our barcoding revealed that most of the resulting sequences (~80%) were PCR duplicates (vs ~20% or less, for *B2m*, *Anxa5* and *Cybb*, which are comparatively much more highly expressed). This validates our barcoding approach, and also indicates that poorly covered transcripts (by scRNA-seq) are truly representative of very few mRNA molecules, exacerbating issues of sampling, especially in the context of editing. Thus, our analysis must be strictly limited (for technical reasons) to well-covered transcripts. We can add a line noting this within the discussion and now provide these data as a figure for the reviewers, at the end of this rebuttal.

The statistical model is an important component of this work. More detailed and quantitative evaluation of its performance is needed. For example, how accurate can the model estimate variance v ? In the simulation analysis, does the estimated v correspond to expected v values accurately? If the simulated editing rates are in 0-5% and 95-100%, why is the estimated v much lower than expected? The cases of editing rates being in 0-5% and 95-100% are extreme cases of high variability, what about more realistic cases such as editing rates being 20%-50%? Indeed, different ranges of editing rates should be simulated to evaluate the model performance.

We thank the reviewer for this comment - we have now added simulations with artificial data testing editing at more physiological ranges (0-50%, comparing multiple ranges) and compared the variances (Figure S5). We report that these correspond well to observed differences in editing rates.

Just for testing purpose, can the authors analyze allelic ratio of heterozygous SNPs (expected to be 1:1) in their data, and determine if variance estimate is small as expected?

In an attempt to test the model on an experimental condition where there should be no variance in editing rate, we looked for a heterozygous genomic SNP from the RNA-seq data pooled from the 24 single macrophages sequenced, as they were all taken from the same mouse. We screened the pooled RNA-seq data for genomic biallelic variants annotated as common and coding in dbSNP 138, and found that the mouse sequenced was potentially heterozygous at 47 sites (i.e. expressed both the annotated reference and alternate base, with the alternate base occurring 15 to 85% of the time at the coordinate of interest). However, coverage across the 24 cells for these 47 sites was poor and the cells exhibited expression from only one of the two alleles for most, if not all, of the sites, consistent with previous reports (for example, Borel et al. AJHG 2015, PMID: 25557783 and Gimelbrant et al. Science 2007, PMID: 18006746). Therefore, variance is actually not low in this situation as expected, so this is not a useful control. We have now added this paragraph to the discussion section.

There seems to be unexplored values of the modeling approach. It is not surprising that editing rate variability exists across cells, and the level of variability may differ for different editing sites. To strengthen the aspect of biological relevance of this work (which is rather weak in the current version), can the authors extend the models to determine, for a given site, whether it is under certain regulation for low (or high) cell-to-cell variability in editing rate? In other words, is it possible to determine whether the variability of an editing site demonstrated in the data is significantly higher or lower than expected by a non-regulated statistical model? This type of modeling extensions may add significant value to this work.

The reviewer appeared to be asking whether the model can be used to demonstrate that under certain conditions, a specific site would be expected to exhibit higher (or lower) variability in editing rate. To address this question, we turned to recently published single cell RNA-seq datasets from bone marrow derived dendritic cells (BMDMs) (Shalek et al, 2015). These datasets have been utilized exhaustively to argue that while cell populations are transcriptionally and genomically homogeneous at the start, they quickly separate into two groups in response to LPS stimulation: a group of “early responders” and a group that eventually catches up, before they merge again later in the response. Our editing analysis of those datasets is presented in Figure 4 (and in more detail in supplemental Figures S9-11). Indeed, as the reviewer hypothesized, the model we constructed when applied to the published datasets indicates changes in variance (per specific site) during stimulation. The biological relevance of this regulation is unclear at the moment, but it is not impossible to imagine that editing of a specific transcript site which occurs in a small subset of cells (a good example is *Cybb*) at time 0, might “pre-determine” the fate of those cells (e.g. toward becoming early responders for example) with the rest of the population catching up soon thereafter.

We thank the reviewer for their insight – these data are in Figure 4 (and in expanded form in Figures S9-11, and have replaced old Figure 3 (showing BMDM derived plots for *Cd36* and *Serinc1*). The comments above are now part of the Discussion.

In the RT-PCR validation, it is intriguing to observe C-to-U changes that are only in single-cells, not in bulk. Are these real C-to-U editing sites? Are they associated with sequence or structural features characteristic of true C-to-U editing? How many other types of nucleotide changes were observed for each experiment? Could these single-cell-specific changes be caused by RT mistakes, or other types of artifacts?

We frequently observe in our Sanger sequencing data additional sites of C-to-U editing neighboring sites that were identified bioinformatically from the bulk data. This is the case from single cells as well as at the population level. Often, these are sites which, given our strict detection criteria, would not rise past our limits for detection (5% minimum editing rate) - however, we do think that they are real. They are unlikely to be RT mistakes, since (as noted in our prior response to reviewer#2), (a) reverse transcriptase (RT) has characteristic features (which include error rate - roughly 1/1,500 to 1/30,000 bases (Roberts et al. Science 1988) and also preferred misincorporation profiles that are very different from what we observe and (b) we observe recurrent mismatches from the reference, that are C-to-T changes and occur at distinct transcripts (with non-identical barcodes - Figure 3, Figure S8a). The likelihood that RT error (and not enzymatic deamination) would give rise to such consistent nucleotide transitions at similar locations within different transcripts is negligible.

To provide more convincing validation data, the authors should determine whether the variation (and editing rates) observed in RT-PCR faithfully reproduced what they observed in the sequencing data combined with statistical modeling. That is, a more quantitative assessment of the validation data is needed, which may necessitate include of more than 8 cells for each site.

The reviewer notes that better quantitation might necessitate providing additional cells per site. Noting that the single cell amplicon data do recapitulate adequately editing at specific coordinates, we nevertheless augmented the number of cells queried (these data are provided in new supplemental Figure S8b). We hope that this will adequately address reviewer concerns.

There are quite a number of typos in the text that need to be corrected. In addition, technical parts of the writing are sometimes obscure. For example, what is "p value that is a fraction of 0.05"? Rather than stating it in this way, a specific p value or p value range should be given.

We thank the reviewer - we hope and expect that the revised version contains no typos or obscurities. In the specific example the reviewer mentions, we have replaced the expression "p value that is a fraction of 0.05" with "p value is less or equal to 0.05".

Reviewer #4 (Remarks to the Author):

The manuscript entitled "RNA editing generates sequence diversity within cell populations." by Harjanto et al is dealing with the intriguing issue of exploring RNA editing at the single cell data. The question this paper is dealing with, touches one of the most fundamental aspects of biological diversity, one that created by post transcriptional modifications at the level of single cell. The present picture about these modifications is clearly far from the real biological states and masked by the current methods that averaged millions of cells into one measurement. The recent rapid development of single cell sequencing has the potential to reveal the true nature of post-transcriptional modifications. For example, when we see RNA editing at 20% ratio at a site in a tissue level, with standard RNAseq technique, is it mean that all cells have 20% editing or it means that some cells has 100% at the site and other has 0%? As far as I know, this paper is the first work that touches this fundamental issue, focusing mainly on C-to-U RNA editing. The paper is interesting, deals with an important subject and the authors clearly present the motivation for the project. I was a referee of the same paper at an earlier version in other journal. At this current version the author address all the concerns I raised there and I believe that this version is much better and I have no additional comments.

We thank the reviewer for their interest and comments.

Figure Legend for the accompanying figure:

Figure for the reviewers: analysis of editing within single cells must be strictly limited (for technical reasons) to well-covered transcripts. a) Elimination of PCR duplicates from analysis (the Rac1 transcript 3'UTR is used as an example). Left: Screenshot of subclone sequences aligned to the reference genome, unfiltered for PCR duplicates. Right: collapsed PCR duplicates using 6nt barcodes. Each barcode represents a unique mRNA molecule in the original cell. b) Percentage of unique reads per cell. The high amount of PCR duplication is indicative of low complexity in the original sample.

a**b**

Cell #	Unique barcodes
1	13%
2	20%
3	37%
4	20%
5	23%
6	23%
7	33%
8	40%
9	26%
10	33%
11	50%
12	20%
13	37%
14	23%
15	10%

Response to Reviewers' comments:

Reviewer #1 (Remarks to the Author):

Original concerns not addressed and thus opinion not changed.

We thank the reviewer for their interest and comments.

Reviewer #2 (Remarks to the Author):

The authors have addressed my comments regarding the possibility of polymorphisms and errors incurred during reverse transcription. As far as the title goes, I would avoid claiming that the population of cells is "otherwise homogeneous" and go with something closer to the first option that was proposed in response to comment 4.

We are glad that we have addressed the reviewer's comments to their satisfaction. We have provided two possible titles for the manuscript (listed at the top of the main text) - we leave the final choice of title to the discretion of the editor.

Reviewer #3 (Remarks to the Author):

NA

We thank the reviewer for their interest and comments.